# Phenolic Compounds from New Natural Sources—Plant Genotype and Ontogenetic Variation

**DOI:** 10.3390/molecules28041731

**Published:** 2023-02-11

**Authors:** Renata Nurzyńska-Wierdak

**Affiliations:** Department of Vegetable and Herb Crops, Faculty of Horticulture and Landscape Architecture, University of Life Sciences in Lublin, Doświadczalna 50a, 20-280 Lublin, Poland; renata.nurzynska@up.lublin.pl

**Keywords:** plant development, secondary metabolites, variability factors, therapeutic effect, antioxidant activity

## Abstract

Phenolic compounds (PCs) are widespread secondary metabolites with potent biological activity. Their sources are mainly plants from cultivated and natural states, providing valuable protective and health-promoting extracts. The wide biological activity of PCs (antioxidant, anti-inflammatory, antimicrobial, antiatherosclerotic, antidiabetic, antiallergic, prebiotic, antimutagenic) means that new sources of PCs are constantly being sought, as exemplified by extracting these compounds from tissue culture or agricultural by-products. Plant phenols show marked qualitative and quantitative variation not only at different genetic levels (between and within species and clones) but also between different physiological and developmental stages. Assessing genetic and seasonal variations in phenolic content and activity allows for selecting the best time to harvest the plant. Learning about the causes of PCs’ variability and putting this knowledge into practice can significantly increase PCs’ yields and extract the most valuable compounds. The health-promoting properties resulting from consuming products rich in plant PCs are undeniable, so it is worth promoting high-phenolic products as a regular diet. This paper presents an overview of different sources of PCs for use as potential therapeutic alternatives. Additionally, factors of variation in the phenolic complex at the genome and ontogeny levels, relevant in practical terms and as a basis for further scientific research, are presented.

## 1. Introduction

Phenolic compounds (PCs) are essential components of food. In addition to their strong antioxidant properties, they influence the sensory characteristics of food products [1,2]. They are the most widely distributed secondary metabolites, ubiquitous in the plant kingdom, and rarely found in bacteria, fungi and algae [1,2,3]. PCs are stress metabolites produced by plants to protect themselves against abiotic and biotic factors [3]. Structurally, PCs are secondary plant metabolites characterised by at least one aromatic ring with one or more attached hydroxyl groups. The latter are called polyphenols [3,4]. Based on their structure (number of aromatic rings and the way they are connected), PCs are divided into five subgroups: (1) phenolic acids (hydroxybenzoic and hydroxycinnamic acids), (2) flavonoids (flavonols, flavan-3-ols, flavones, flavanones, isoflavones, flavanones and anthocyanidins), (3) coumarins, (4) lignans and (5) stilbenes [5]. Two main pathways are involved in the biosynthesis of PCs, the shikimic acid pathway and the malonic acid pathway, with the former being the main pathway in plants [1]. The chemical structure of PCs can affect their bioavailability and biological activity. An example is the slight differences in the chemical structures of resveratrol, piceatannol and pterostilbene. Although they are chemical analogues, piceatannol and pterostilbene have higher activity, probably due to their higher resistance to intestinal and hepatic metabolism, resulting from differences in the number of hydroxyl and methoxyl groups. It is hypothesised that the different antioxidant activities of PCs may be related to their ability to act as radical scavengers and are due to their chemical structure [3,4].

The presence of PCs in the diet is crucial in preventing many diseases of civilisation. Research demonstrates the beneficial effects of phenols on the human body, including remarkable antioxidant, vasoactive and anti-inflammatory effects, and the ability to interact with enzymes and cell receptors [6,7,8]. The quantitative and qualitative composition of the polyphenolic fraction determines fresh plant materials’ sensory quality and biological value. The health benefits of consuming phenols largely depend on their bioaccessibility and bioavailability in the digestive tract and circulatory system [6]. The bioactivity of PCs tends to be stronger or weaker, affecting different systems and tissues. Polyphenols are thought to have low bioavailability due to interactions with the food matrix and metabolic processes mediated by the liver, gut and microflora [9].

The bioavailability of phenols widely varies, and the most abundant phenols in the diet are not necessarily the most active in vivo due to poor absorption, rapid excretion, high metabolism and lower intrinsic activity, or all of them [10]. The low bioavailability of PCs has been recognised as a limitation to using these compounds as practical tools in treating certain diseases. However, it should be added that the biological activity of PCs may be mediated by their metabolites, which are produced in vivo, as indicated by recent studies confirming their antioxidant and anti-inflammatory properties [9]. The biological activity of PCs and the potential activity of their metabolites is still a topic of research [4]. For these reasons, new sources of PCs are being sought, and individual phenolic complexes are being analysed, paying attention to changes in their content in plants, raw materials and plant products. Bioavailability rates largely depend on the maturity stage of samples, the genotype and environmental conditions, which directly affect the phenolic content, solubility, chemical structure and biological activity [11]. In the following section, recent findings on the content of PCs in commercial plants are presented, considering new sources of polyphenols, as well as changes in PCs’ levels and activity as influenced by genetic and ontogenetic factors.

## 2. New Sources of Phenolic Compounds

Today, there is a growing demand from the food, pharmaceutical and cosmetic industries for plants with exceptional metabolic properties, including antioxidant properties and PC-rich plants. These plants are obtained from two sources: cultivation and the natural state. Polyphenols are isolated and purified from plants (fruits, vegetables and agricultural by-products) and converted into medicines and supplements [4,5,8]. The raw material base of cultivated plants is often limited or not of interest in this aspect. However, researchers are increasingly interested in wild flora [12,13,14,15], as well as plants previously used in other areas [16]. The production of metabolites in vitro using metabolic engineering to enhance the bioactivity and stability of secondary plant metabolites is also gaining importance [17].

### 2.1. Wild-Growing Plants

Wild plants are still a readily available but not fully explored source of bioactive compounds with potential phytotherapeutic benefits. The results of selected studies on the phenolic profile of wild-growing plants are presented below. *Ageratina petiolaris* (Moc. and Sessé ex DC.) RM King & H. Rob. (Asteraceae) is widespread in Mexico and widely used in traditional medicine for digestive disorders, indigestion, kidney disease, rheumatism and nervous disorders, among others [18,19]. Pérez-Ochoa et al. [19] reported that *A. petiolaris* samples collected in situ have higher PC contents (Table 1) than plants grown ex situ. Interestingly, the production of gallic acid and rutin was not sufficiently detected in the cultivated samples and was found in the samples from the natural state. It indicates a possible plant response mechanism in modifying biosynthetic processes related to stress conditions in the growing environment. *Rumex* L., the second largest genus in the Polygonaceae family with more than 200 species, is mainly found in the northern temperate zone. Species of the genus are valued in worldwide folk medicine, e.g., in South Africa, America, India, China and Turkey [20]. Feduraev et al. [21] showed a positive association of antioxidant activity with the total content of PCs (r = 0.785–0.921, *p* ≤ 0.05), flavonoids (r = 0.602–0.918, *p* ≤ 0.05), proanthocyanidins (r = 0.721–0.842, *p* ≤ 0.05) and tannins (r = 0.591–0.776, *p* ≤ 0.05) in the leaves of wild plants of the genus *Rumex*. The results of the study identified the most promising species for pharmaceutical and food use, distinguished by their high levels of relevant phenolic compounds: *R. maritimus*—flavonoids, *R. acetosella*—hydroxycinnamic acids, *R. sanguineus*—catechins, *R. sanguineus*, *R. obtusifolius*, *R. crispus*—proanthocyanidins and *R. obtusifolius*—tannins. The genus *Sideritis*, in the family Lamiaceae, includes herbaceous plants growing mainly on the Mediterranean coast, where 140 native species are known. Many of them are used to make traditional aromatic herbal teas that effectively prevent and treat various diseases. The biological activities of *Sideritis* herb extracts, mainly antioxidant and antimicrobial, are related to the high content of total phenols and chlorogenic acid [22,23]. Bardakci et al. [24] assessed the phytochemical composition and antioxidant potential of *S. congesta* extracts and found the presence of 22 active phenolic metabolites (Table 1). The ethyl acetate fraction had the highest phenolic compound content (227 ± 5 mg∙g^−1^ gallic acid equivalent (GAE)) and antioxidant activity, as well as the proportion of verbascoside (48%) and martinoside (2.96%). Another example is *Valeriana carnosa* (Sm.) Dufr. (Caprifoliaceae), commonly known in Argentina as ‘Ñamkulawen’ or ‘the medicine that cures seven diseases’. The roots and rhizomes of this plant are traditionally used in decoctions to treat various conditions, including liver, respiratory, circulatory, urinary and digestive conditions, and have analgesic, anti-inflammatory, anticancer, antidepressant and wound-healing properties [13,25]. Guajardo et al. [13] identified eighteen PCs in *V. carnosa* root extracts (Table 1). A significant positive correlation was found between antioxidant activity and PC content, as determined using the stable radical 2,2-diphenyl-1-picrylhydrazyl (DPPH).

*Vaccinium macrocarpon* Aiton (Ericaceae) is a North American species of cranberry in the genus *Vaccinium*, subgenus *Oxycoccus*. The plant occurs naturally in North America’s eastern and central regions but is increasingly cultivated in Europe and other continents [26,27]. The fruits of *V. macrocarpon* are valued for the variety of biologically active compounds they contain, and they accumulate large amounts of PCs (Table 1), organic acids and mineral substances. In cranberry fruit, PCs are one of the dominant groups of biologically active compounds with pronounced biological effects [26,27]. Evaluation of fruit samples of American cranberry varieties grown in Lithuania showed that the total PCs’ content ranged from 10.61 to 18.06 mg GAE∙g^−1^ dry matter (DM) [27]. Narwojsz et al. [26] confirmed the variable levels of PCs in the fruits of different cranberry cultivars by indicating that the antioxidant activity of fruit extracts was correlated with the content of PCs, flavonoids and proanthocyanidins, while antitrypsin activity was correlated with the content of PCs and anthocyanins. Woody plants play a significant role in nature and the human economy. Many of them are known as the medicinal species. One of the most important are plants of the genus *Pinus*, whose medicinal qualities are mainly related to the presence of essential oils. Pine extracts contain many polyphenols; nevertheless, individual compounds are characterised by different concentrations, types and levels of their bioactivity [28]. Fkiri et al. [29] showed significant amounts of PCs and high levels of antioxidant activity in *P. nigra* Arn needle extracts (Table 1). Significant differences were found between the subspecies and origin, indicating the importance of genetic and environmental factors for PC levels and antioxidant activity. Kovalikova et al. [30] showed promising but variable phenolic content and antioxidant potential of extracts prepared from buds of different trees, *Acer pseudoplatanus* L., *Betula pendula* Roth and *Quercus robur* L., grown under different environmental conditions. DPPH radical scavenging was effective for all tested extracts (80–95% depending on species, extract type and assay). The authors indicate that the antioxidant activity of the tested extracts may be due to variability in the composition, content and chemical nature of the different active compounds and synergies between them and other natural substances. Interspecies differences regarding variations in phenolic composition and biological activity were also confirmed by Saada et al. [31], who analysed the aboveground parts of six Tunisian wild medicinal plant species (*Retama raetam*, *Nitraria retusa*, *Pituranthos tortuosus, Zygophyllum album*, *Juncus maritimus* and *Rubia timctorum*). The species *R. raetam* (Fabaceae) proved to be the richest in polyphenols and flavonoids (23.93 mg GAE∙g^−1^ DM and 3.58 mg quercetin equivalent (QE)∙g^−1^ DM, respectively), showing the best antiradical activity in DPPH and ferric reducing ability of plasma (FRAP) assays and antimicrobial activity. Simultaneously, the extract from *J. maritimus* (Juncaceae) was much more effective in terms of β-carotene oxidation inhibition. In contrast, *R. tinctoria* (Rubiaceae) plants accumulated the most condensed tannins (2.41 mg catechin equivalent g^−1^ DM).

**Table 1 molecules-28-01731-t001:** Selected examples of phenolic compounds found in wild plants described above.

Compound	Plant Species	Ref.
Apigenin-7-glucoside, gallic acid, luteolin-7-glucoside, ρ-coumaric acid, robinin, rosmarinic acid, rutin	*Ageratina petiolaris* (Moc. and Sessé ex DC.) RM King & H. Rob. (Asteraceae)	[19]
Flavonoids, phenols, tanins	*Pinus nigra* Arn. ((*Pinaceae*)	[29]
Catechins, flavonoids, hydroxycinnamic acids, proanthocyanidins, tannins	*Rumex* spp. (Polygonaceae)	[21]
Chlorogenic acid, leucoseptoside A, martinoside, verbascoside	*Sideritis* spp. (Lamiaceae)	[22,24]
Chlorogenic acid, ferulic acid, gallic acid	*Valeriana carnosa* (Sm.) Dufr. (Caprifoliaceae)	[25]
Anthocyanins, flavan-3-ols, flavonols, phenolic acids	*Vaccinium macrocarpon* Aiton (Ericaceae)	[26,27]

Wild plants represent a valuable genetic resource that can be used in breeding programmes to increase the resistance of crop plants and improve their nutritional and pharmacological value [17,21,22,23]. Functional foods of plant origin have gradually become an area of growing research interest due to their health benefits for the human body, and plant polyphenols have gained widespread attention as some of the most prevalent chemical constituents. In addition to naturally occurring plants, cultivated plants, such as cereals, are also an interesting source of polyphenols and are increasingly being studied for their PCs and antioxidant activity [32,33,34].

### 2.2. Cereals

Growing nutrition and health awareness results in an increased effort to develop functional food products with bioactive ingredients. Cereals are an essential food category for many of the world’s populations, with an annual production of more than 2700 tonnes when supply and demand are balanced [32]. Whole-grain products are a vital food category in the human diet and are an invaluable source of carbohydrates, proteins, fibre, phytochemicals, minerals and vitamins. The tannins, phytoestrogens and other PCs in cereal grains (Table 2) show valuable health-promoting activities: antioxidant, immunomodulatory, cardioprotective and anticancer activities [32,33,34].

It has been proven that pigmented cereal varieties are sources of bioaccessible polyphenols with antioxidant activity [34]. *Sorghum* (*Sorghum bicolor* (L.) Moench) and pearl millet (*Pennisetum glaucum* (L.) R. Br.) are cereal crops from the Poaceae family cultivated in various regions of the world [33,34]. *S. bicolor* is the world’s fifth most important cereal crop after wheat, rice, maize and barley. The species is distinguished, among other things, by its strong stress tolerance and high genetic diversity. Cultivations include sorghum varieties with different grain colours (white, yellow, red, brown), which is a genetically determined trait. Sorghum can also be a valuable PC source with potential importance in producing antioxidant extracts [33,34]. Bhukya et al. [34] presented a comparison of differences in the total phenolic content and antioxidant capacity of grains of different sorghum varieties, which ranged from 575.05 to 3161.87 mg GAE∙kg^−1^, 888.33 to 4230.14 mg GAE∙kg^−1^ and 1274.91 to 2885.72 mg GAE∙kg^−1^ in white, red and brown sorghum genotypes, respectively. Eleven different phenolic acids were identified, of which ferulic acid had high expression only in the white and red sorghum genotypes. Red sorghum grains showed high phenolic and antioxidant activities in vitro. Similarly, Mawouma et al. [38], comparing different varieties of sorghum and millet, found that red sorghum contained the highest amounts of total polyphenols (82.22 mg GAE∙g^−1^ DM), total flavonoids (23.82 mg QE∙g^−1^ DM) and total 3-deoxyanthocyanidin (9.06 mg∙g^−1^ DM). The highest antioxidant activity against DPPH was determined in yellow-pale sorghum (87.71%), followed by red sorghum (81.15%). The above study [38] found higher amounts of PCs in sorghum than in pearl millet. Polyphenols are the main bioactive compounds of sorghum and are present in all varieties of this cereal crop. It should be added that polyphenols and phytates in sorghum and millet grains are also known to be anti-nutritional factors, as they form insoluble complexes with minerals such as iron, zinc and calcium, reducing their bioavailability. The nutritional value of cereal grains is, therefore, in some contradiction with the antioxidant value associated with polyphenol levels. Mawouma et al. [38] showed that white sorghum has the lowest levels of copper, iron and zinc and low levels of polyphenols. On the other hand, red sorghum is the richest in polyphenols, flavonoids and 3-deoxyanthocyanidin, and was found to have the lowest phytate content. Yellow- pale sorghum showed the highest antioxidant activity (93.14%), followed by red sorghum (86.43%). It can be assumed that the individual sorghum varieties differ not only in their nutritional value but also in their antioxidant potential, regulated, among other things, by their polyphenol content [33,34]. Maize (*Zea mays* L.) is a cereal crop widely distributed and consumed worldwide, especially in developing countries. Purple (violet) maize, native to the Andes region of present-day Peru, is nutritionally and chemically similar to the more commonly consumed yellow maize. It is rich in starch, non-starch polysaccharides, proteins, lipids, minerals and vitamins. What distinguishes purple maize is the presence of anthocyanins and other PCs with valuable health-promoting activities [39,40]. Ramos-Escudero et al. [37] showed that purple maize extract contains phenolic acids, which exhibited significant in vitro antioxidant activity, which correlated well with decreased malondialdehyde (MDA) formation and increased activity of endogenous antioxidant enzymes. The tested extract was shown to be capable of significantly reducing lipid peroxidation (reduction in MDA concentration) and simultaneously increasing the activity of endogenous antioxidant enzymes (catalase, total peroxidase and superoxide dismutase) in organs isolated from mice (kidney, liver and brain).

Rice (*Oryza sativa* L.) is a significant cereal for more than half of the world’s population, especially in Asia, as it is a significant source of carbohydrates and other nutrients. Summpunn et al. [41] showed that rice’s antioxidant components and activity are genetically determined, with white rice having the lowest total phenolic content. Rice grass had the highest total phenolic content (the exception was the Yar Ko variety, where the highest total phenolic content was found in brown rice). The authors of this study conclude that, depending on the rice variety, processing into brown rice, sprouted brown rice or rice grass contributes to increased antioxidant content and power. In contrast, processed white rice decreased antioxidant compounds (PCs, ascorbic acid, carotenoids and γ-oryzanol) as well as antioxidant activity. The results of this research are significant and promising in terms of producing rice as a functional ingredient.

### 2.3. Plant Cell, Tissue and Organ Cultures (PCTOCs)

PCTOCs can produce and accumulate many medically valuable secondary metabolites. However, few metabolites are biosynthesised using plant platforms, except for the natural pigment anthocyanin. Currently, PCTOCs can produce phenols (Table 3), alkaloids, terpenes and steroids [42]. Many plants containing high-value PCs are difficult to cultivate on a large scale due to specific environmental requirements [43,44]. Suitable PCTOCs have become an attractive alternative PC source, especially when natural resources are limited [43,44]. The use of PCTOCs can be advantageous because they grow under strictly controlled conditions, allowing the easy addition of hormones, biosynthetic precursors and other compounds [45,46,47,48]. Additionally, these methods are often more efficient and cost-effective than other methods of obtaining active substances [43,49].

Depending on the species, rosmarinic acid yields from suspension cultures can exceed 10% of cell weight and 6 g∙L^−1^ of a medium, and *Catharanthus roseus* and *Camptotheca acuminata*, species used in medicine for their anticancer alkaloids, produce significant amounts of anthocyanins (>200 μg∙g^−1^ fresh matter (FM)) in cell culture [43]. Yang et al. [49] found that it takes at least 3 years to harvest *Glycyrrhiza inflata* plants for medicinal purposes and that flavonoid productivity from suspension cell cultures grown for 21 days was higher than that of a 3-year-old plant. Naik and Al.-Khayri [50] described the production of polyphenols from the suspension culture of date palm cells (*Phoenix dactylifera* L., cv. Shaishi) and found the highest production of biomass (62.9 g∙L^−1^ FM and 7.6 g∙L^−1^ DM) and polyphenols (catechin—155.9 µg∙L^−1^, caffeic acid—162.7 µg∙L^−1^, kaempferol—89.7 μg∙L^−1^ and apigenin—242.7 μg∙L^−1^) in an 11-week culture. Furthermore, a study by Devrnja et al. [55] showed that the chemical profiles of methanolic extracts from the shoots and roots of in vitro-cultivated tansy (*Tanacetum vulgare* L.) are quantitatively and qualitatively different from those obtained from wild plants. Methanolic extracts from in vitro-cultured roots were richest in 3,5-dicaquioylquinic acid, the concentration of which was 6-fold higher (10,220 mg∙g^−1^ DM) than in the extract extracted from the roots of wild rampion (1.684 mg∙g^−1^ DM).

In a study by Suvanto et al. [58], 18 cell suspension cultures from 12 plant species from Finland, Sweden and Norway were evaluated for their polyphenol synthesis capacity and suitability for future research and applications. The plant species represented eight different genera and four families (Rosaceae, Ericaceae, Caprifoliaceae and Poaceae): *Rubus chamaemorus* L., *R. idaeus L*., *R. arcticus L., R. saxatilis* L., *Fragaria* × *ananassa* (Duchesne ex Weston) Duschesne ex Rozier ‘Senga Sengana’, *Sorbus aucuparia* L., *Vaccinium myrtillus* L., *V. vitis-idaea* L., *Empetrum nigrum* L., *Lonicera caerulea L.* var. *kamtschatica*, *Avena sativa L*. and *Hordeum vulgare* L. Cultures of Rosaceae were the most efficient in producing hydrolysable tannins but produced few proanthocyanidins. Cultures of Ericaceae were the most efficient in producing proanthocyanidins [57,58]; in the case of *V. myrtillus,* they were comparable even to fruit and leaves. Several of the cell cultures tested showed the ability to produce a wide variety of polyphenols, including high-molecular-weight tannins, which gives hope for further studies on, for example, the accumulation of specific polyphenols or the biosynthesis of polyphenols in the culture cell. Many of the cell suspension cultures studied showed a relatively low polyphenol content compared to plants. The most notable exceptions to this rule were *S. aucuparia*, *V. myrtillus* and *E. nigrum*, which yielded a complex set of different PCs, including oligomeric tannins. Anthocyanins were detected in several species that occur naturally, but their concentrations were lower than those in the fruit [58].

Recently, polyphenols have been investigated for their potential use in cancer therapy. Polyphenols, as natural antioxidants, inhibit the oxidation of lipids, proteins and nucleic acids, thus preventing the initiation of oxidative chain reactions. These compounds can significantly reduce the risk and incidence of cancer by scavenging unstable molecules that could initiate carcinogenicity [59,60,61,62]. In vitro plant cell cultures could serve as a good source of PCs with different cytotoxic activity towards normal and cancer cells [51]. Callus culture of *Iris pseudacorus* L. (Iridaceae) established from leaves yielded valuable PCs (Table 2) [54]. Under normal conditions, calli accumulated 0.4% DM polyphenols, while the addition of phenylalanine resulted in a 1.5-fold increase in isoflavonoid production, allowing the accumulation of 0.69% polyphenols in callus dry matter. Tectorigenin, a promising chemotherapeutic and chemo-preventive agent for cancer treatment, was produced in large quantities (0.3% DM). Polyphenols sensitise solid tumours to alkylating agents such as cisplatin and induce apoptosis, cell cycle arrest, or both. Combinations of alkylating agents, particularly cisplatin, with polyphenols, show promising anticancer activity in lymphoid leukaemia (apigenin showing the most remarkable effect) [62]. In this respect, the efficient extraction of active polyphenols from PCTOCs seems an appropriate research direction.

### 2.4. Agri-Food By-Products

Food processing causes significant nutrient losses, and waste generation causes severe economic and environmental problems [63,64]. Approximately 89 million tonnes of food waste is generated annually in Europe, which is expected to increase 40-fold in the coming years. The agri-food industry generates large amounts of waste from food production, preparation and consumption. In many cases, these products are increasingly recognised as a valuable and low-cost source of bioactive compounds, including PCs [64,65].

#### 2.4.1. Vegetable and Fruit Processing

Fresh fruits and vegetables represent a significant segment in the functional and nutritional food sector. In fruit and vegetable production and processing, unused parts of the plants remain, often containing significant amounts of bioingredients [64,65,66,67]. Fruits and vegetables include hulls, skins, pods, pomace, seeds and stems, which are usually discarded despite containing potentially beneficial compounds such as carotenoids, dietary fibre, enzymes and polyphenols (Table 4). The use of fruit and vegetable waste in food and pharmaceutical products is gaining importance [64,65,66]. Vaz et al. [65] evaluated vegetable pomace residues (artichoke, red pepper, cucumber and carrot) for juice. Artichoke pomace had a high concentration of total PCs (8340.7 mg∙kg^−1^) compared to red pepper (304.4 mg∙kg^−1^), carrot (217.4 mg∙kg^−1^) and cucumber (195.7 mg∙kg^−1^). In artichoke and carrot pomace, more than 98% of the PCs were phenolic acids, while in cucumber and pepper, the most representative PCs were flavanones. The authors of this study highlight the significant concentration of bioavailable PCs after digestion in the gastrointestinal tract in vitro and the low concentration of PCs after digestion in the colon in vitro, indicating their chemical metabolism by the intestinal microflora. Tomatoes are an essential raw material for processing. The production of juices, sauces, ketchup or purees produces significant quantities of by-products (peel, seeds, small amounts of pulp), increasing the cost of the primary products by 5–30%. The by-products contain PCs, including compounds with significant antioxidant and antimicrobial activities [66]. Blackcurrant buds and leaves are also excellent sources of beneficial PCs. Vagiri et al. [67] presented an optimised extraction of PCs from different parts of the blackcurrant plant, finding 23 compounds in buds, of which 22 were found in fruits and leaves (Table 4).

Apples and the products obtained from them are rich sources of PCs. Apple pomace, a by-product of the apple juice industry, is a good source of polyphenols, minerals and dietary fibre [64,66]. These products contain PCs (289.1 ± 4.2 mg EGA∙100 g^−1^ DW), and among them therapeutically relevant compounds for diabetes, cancer and cardiovascular and neurocognitive diseases (Table 4) [66,71,72,73]. The citrus industry (juices and essential oils) produces large quantities of by-products, such as peels and seed residues, with citrus residues presenting a different composition of polyphenols than apples [63]. Citrus peels are a rich source of natural flavonoids and contain more PCs than their edible parts [74]. The total phenolic content of the peels of lemons, oranges and grapefruits can be 15% higher than in the peeled fruit [75], and similarly, the level of hydroxycinnamic acid is much higher in the peel than in the juice [76]. Therefore, by-products secreted by apple and citrus processing plants are a rich source of PCs. Reusing valuable vegetable and fruit residues is still limited compared to their potential. Further research is needed to find and develop new products and applications for vegetable and fruit processing residues and by-products.

#### 2.4.2. Oil and Wine Production

The oil industry generates many by-products that, when properly treated, can be used in food formulation. Oil and wine production are two major agri-food economic activities in Southern Europe, generating large amounts of solid and liquid waste (e.g., olive and grape pomace, grape stalks, wine lees), which pose a serious environmental problem but are also a potential source of polyphenols [69,70,77]. Worldwide production of olive pomace is estimated to exceed 2.8 million tonnes per year. Olive pomace is rich in many PCs (Table 4) [69,70]. Multescu et al. [78] confirmed that by-products of industrially obtained vegetable oils are a good source of many biological functional compounds, including PCs, especially flavonoids with antioxidant properties. The authors demonstrated the antioxidant activity of extracts from oil by-products, with the highest values for the FRAP method represented by grape seed flour (4716.75 mg Trolox∙g^−1^), followed by sunflower meal (1350.86 mg Trolox∙g^−1^) and rapeseed flour (1034.92 mg Trolox∙g^−1^).

Grape marc, the residue produced during the pressing of grapes, consists of grape skins and seeds. Globally, 9 million tonnes of this waste is produced annually, representing an average of 20% of the total grapes used for wine production [79]. Pomace is an essential source of PCs, mainly anthocyanins, flavonols, flavonoids, phenolic acids and stilbenes [80]. Analysis of the polyphenol complex in the seeds and skins of Grenache, Syrah, Carignan Noir, Mourvèdre, Counoise and Alicante Bouchet grapes and their pomace remaining after vinification by Ky et al. [81] found that seeds contained higher amounts of total polyphenols (up to 44.5 mg GAE∙g^−1^ DM) than extracts from skins (31.6 mg GAE∙g^−1^ DM). The seeds also had the highest antioxidant capacity. It is worth noting that despite the vinification process, the pomace still contained a significant amount of proanthocyanidins and several anthocyanin glycosides. The high antioxidant activity of grape pomace suggests that it is an interesting source of natural antioxidants. In addition, Hamza et al. [82] provided evidence that a grape seed extract with high concentrations of total PCs (141.26 mg GAE∙g^−1^) and flavonoids (68.16 mg QE∙g^−1^) has anticancer effects in liver cancer by inhibiting cell proliferation, inducing apoptosis, modulating oxidative damage and suppressing the inflammatory response.

#### 2.4.3. Other Plant By-Products

Plant seeds are one of the richest sources of polyphenols, accumulated in the seed coat, cotyledons, or both. Cereal by-products, such as bran, are rich in flavonoids and phenolic acids (Table 4), compounds with a significant impact on reducing the incidence of non-communicable diseases [66]. Plants from the Malvaceae and Cannabaceae families are known for their fibre-rich stems used commercially and industrially. Their seeds, often considered a by-product of fibre processing, may prove helpful in combating hypercholesterolemia, diabetes, cancer and oxidative stress. Okra (*Hibiscus esculentus* L.), cotton (*Gossypium* L. spp.), hemp (*Cannabis sativa* L.) and kenaf (*Hibiscus cannabinus* L.) are some examples of promising sources of natural polyphenols [16]. Ong et al. [83] showed that polyphenol-rich okra seed extract potentially protects blood vessels. PCs with scientifically proven health benefits (Table 4) have been found in cottonseed [16]. Defatted kenaf seed meal has considerable potential as a functional food ingredient, with positive health-promoting effects attributed to phenols and saponins. This product has been shown to induce a high protective effect against oxidative stress and inflammation in rats with hypercholesterolemia [84]. Hemp seeds are high in fibre, fat, protein, sugars, organic acids and phenolic acids (Table 4). Hemp seed extracts show antioxidant activity, cytotoxic activity against NCI-H460 cells, antimicrobial activity, especially against *Bacillus cereus, Listeria monocytogenes* and *Enterococcus faecalis*, and, to a lesser extent, antifungal activity [68]. Frassinetti et al. [85] reported that seeds and sprouts of *C. sativa* contain polyphenols and exert beneficial effects on yeast and human cells (antioxidant and antimutagenic effects). Furthermore, Irakli et al. [86] reported that the antioxidant properties of hemp seed extract are mainly due to the presence of PCs rather than other antioxidants (tocopherols and carotenoids). 

The recovery of other parts of the crop not previously harvested as a crop and so-called production waste offers the potential to produce PC-rich products from the entire crop, not just the primary raw material. The recovery of PCs from agri-food waste should be considered an extremely valuable, albeit complex, process involving PCs’ extraction, postprocessing of the resulting extracts, selective separation of mixtures or individual components and final freeze-drying processing steps to produce a powder form [70].

## 3. Plant Genotype and Phenolic Complex

The ability to synthesise active substances more or less intensively is an inherited and genetically determined property. The ability to synthesise PCs has emerged throughout evolution in various plant genotypes, thus enabling plants to cope with ever-changing environmental challenges during evolution [87]. The successful adaptation of some higher members of the class Charophyceae (cluster Algae) to terrestrial conditions has been mainly achieved through the massive formation of ‘phenolic UV screens’ [88]. PCs are widespread in the plant world. However, the distribution of many phenolic structures is restricted to specific genera or families, making them convenient biomarkers for taxonomic studies and relevant for breeding work [1,21].

### 3.1. Genetic Variation vs. Taxonomy and Breeding

It has been shown that there is a diversity of phenolic profiles within the species (Table 5), which proves to be helpful in plant breeding. Bhandari et al. [89] found a relatively higher total flavonoid content and antioxidant activity in new broccoli breeding creations than in a commercial variety. The total PCs’ content was genotype- and developmental stage-dependent and positively correlated with antioxidant activity. This finding identifies the new broccoli breeding lines as more nutritionally and pharmacologically valuable than the commercial variety. Bakhtiar et al. [90] investigated the variability of agromorphological traits, the content of PCs and other components in the aboveground parts of Iranian species of the genus *Trigonella* and related species of *Medicago* L. and searched for distinctive species for further use in breeding and use programmes. Strong correlations were found between some morphological traits and phenolic levels, e.g., stem diameter was related to ellagic acid (*R*^2^ = 0.71, *p* ≤ 0.01), and internode length and petiole length were related to apigenin content (*R*^2^ = 0.76, *p* ≤ 0.01 and *R*^2^ = 0.87, *p* ≤ 0.05, respectively). Two species were selected for further research, breeding work and exploitation in the food and pharmaceutical industries: *T. filipes* and *T. spruneriana*. In contrast, Fkiri et al. [30] showed significant differences in the total phenolic, flavonoid and tannin contents of *Pinus nigra* Arn needles. The total phenols ranged from 15.67 to 47.53 mg GAE∙g^−1^ DM, which also translated into different antioxidant activities (36.08–99.05% DPPH). Similarly, Khorsand et al. [91] showed a significant and positive correlation between rosmarinic acid content and antioxidant activity of extracts from *Origanum vulgare* L. According to the authors of the studies cited here, such natural variability in PCs’ levels offers the opportunity to use elite nutraceutical and pharmaceutical plants and domesticate highly antioxidant genotypes.

Berry crops of the genera *Vaccinium*, *Rubus*, *Ribes* and *Fragaria* are rich sources of phenolic-derived antioxidants. Gudžinskaitė et al. [27] showed high variation in PCs’ levels in the fruit of seven *V. macrocarpon* genotypes, with some cultivars (Woolman, Holliston, Pilgrim, Searles) having different quantitative PC contents and others (Baiwfay, Drever, Bain, Bergman) having similar phytochemical profiles. The total phenolic acids (PAs) content of American cranberry fruit samples ranged from 3.48 (Pilgrim) to 10.68 mg∙g^−1^ (Bergman). The antiradical activity of cranberry fruit extracts depended on the cultivar and the analysis method (ABTS^•+^, TFPH, FRAP, CUPRAC). The Baiwfay and Bergman cvs were outstanding in this respect, and the Pilgrim cv was the least active, which may be related in some way to the quantitative and qualitative proportion of flavonoids. Palmieri et al. [92] reported that the *Fragaria* × *ananassa* Duch. genotype contributes to PC levels under different climatic conditions. More than 50 compounds belonging to 10 different polyphenolic groups were analysed in 9 strawberry cultivars, noting both varietal and environmental contributions. As a result, variable and stable strawberry varieties under different environmental conditions were selected. Significant differences in most varieties were found for benzoic acid derivatives, phenylpropanoids, flavones, flavonols and ellagitannins. Similarly, Lebedev et al. [93] evaluated 25 new breeding lines and standard raspberry cultivars for polyphenol content and antioxidant capacity. They found that fruit antioxidant activity correlated better with the content of total phenols (0.88 and 0.92) and flavonoids (0.76 and 0.88) than anthocyanins (0.37 and 0.66).

The results obtained from the various studies on the variable amounts of PCs in different genotypes may prove extremely useful for the selection of edible or medicinal plant varieties to obtain antioxidant-rich creations, as well as for assessing the quality of plant raw materials and phytoproducts and can also allow the prediction of the antioxidant activity of in vivo extracts. Possible correlations between quantitative and qualitative PCs and antioxidant activity [27,92,93] should be noted, which may prove helpful in developing targeted therapeutic formulations.

### 3.2. Polyploidisation and Biochemical Modifications

One reason for changes in the content of active substances may be an increase in the number of chromosomes (the phenomenon of polyploidisation), occurring spontaneously, under natural conditions, and artificially induced by mutagenic agents. Polyploidy is the possession of three or more complete sets of chromosomes. Genome duplication and its possible adaptive advantages have been essential factors in the speciation and evolution of eukaryotes. Recent studies have shown that autopolyploidisation can remodel the transcriptome and metabolome, generating genomic stress. Consequently, polyploidisation can also result in various molecular and physiological modifications [94]. It is supported by the findings of Asensio et al. [95] on the variability of phenolic compounds in 42 wild populations of bearberry (*Arctostaphylos uva-ursi* (L.) Spreng.) in its natural distribution in the Iberian Peninsula. The methanolic extracts obtained showed a wide range of variation in total phenolic content, with different phenolic profiles, concerning, inter alia, the arbutin level. Moderate levels of variation in genome size—assessed by flow cytometry—and two regions of plastid DNA were detected in the bearberry populations studied. Population genetic and cytogenetic diversity was weak but significantly associated with phytochemical diversity. Based on the study, elite bearberry genotypes with higher antioxidant capacity were identified. Herbal raw materials containing arbutin are used to treat urinary tract infections, chronic cystitis and kidney stones. Bearberry leaf, a natural antioxidant with a high content of phenolic glycosides, mainly arbutin, is listed in many national pharmacopoeias as a urinary tract disinfectant [95,96,97]. The identification of a high-phenolic bearberry genotype offers excellent prospects for breeding and herbal production, especially as the content of the main active ingredient, arbutin, did not vary significantly between the years of the study, which may suggest a more significant influence of genetic rather than environmental variation.

## 4. Ontogenetic Variation

The variability of the chemical composition, and thus the pharmacological action of plant-origin raw materials, is determined by several factors—the chemical profile of a plant changes during ontogenesis. During development, numerous biochemical transformations occur in the plant, resulting in weight gain and the accumulation of biologically active substances. There is a close relationship between the stage of ontogeny and the content of certain active substances in plant organs. This phenomenon is referred to as ontogenetic variability [98,99]. Plants are exposed to various destructive factors during ontogenies, such as attack by herbivores, pathogens and pests, excessive UV radiation, droughts or excessive rainfall and nutrient deficiencies, making it necessary for them to continuously produce and transport appropriate protective metabolites to various organs. Plant phenols play a crucial role as protective compounds when environmental stresses can lead to increased production of free radicals and other oxidative species in plants [1]. Plant ontogeny can influence the biosynthesis and bioaccumulation of various leaf defence metabolites in opposite directions. Goodger et al. [87] showed that total phenolic concentrations decreased significantly, while total terpenoids increased significantly with ontogeny in *Eucalyptus froggattii*. Such variable ontogenetic trajectories may be due to plants maintaining an optimal equilibrium.

### 4.1. Age of the Plant

The age of the plant may be an essential factor in the variability of the chemical composition. The genus *Mentha* spp. (Lamiaceae) includes oil plants containing many valuable PCs, such as flavonoids, phenolic acids or tannins, which are involved in pharmacological properties. Agrotechnical recommendations indicate that cultivation of mint at one site should only be carried out for 2–3 years, but no longer than 4 years, due to deteriorating health and reduced raw material and oil yield. Fialová et al. [100] investigated different mint species and showed high levels of active PCs, especially in 3- and 4-year-old plants, suggesting that a decrease in PCs does not accompany the decrease in essential oil in the 4th year; in contrast, in some cases, the phenolic content increases. Another example is the species *Agastache rugosa* (Fisch. & C.A.Mey.) O. Kuntze (Lamiaceae), in which thirty-six PCs were found [101]. The content of the main compounds (rosmarinic acid, apigenin glucoside, chlorogenic acid) significantly differed between plants of different ages and their mother plant. There was also a close relationship between the annual plant groups and their respective progenitor plants, suggesting an intergenerational phenomenon. Such information may be useful in practice, for example for recommendations regarding the harvesting technology.

### 4.2. Plant Organs

Changes in the content of active compounds and the weight of the plant do not run in parallel but have their own rhythm and dynamics. Environmental factors may accelerate or retard the transition of a plant through particular developmental phases, but they cannot alter the relationship between morphological characteristics and the level of active substances in the organism. Liu et al. [14] studied plants of *Sinopodophylum hexandrum* (Royle) T. S. Ying and found significant differences in PCs’ content, total flavonoids and antioxidant activity between different organs: rhizome > root > fruit > flower > leaf > stem > leaf petiole. It was shown that there is a positive correlation between PCs’ content and antioxidant activity. The rhizome, characterised by a high PC content and potent antioxidant activity, is recommended as a significant medicinal raw material, while the root is recommended as an alternative raw material for natural antioxidants in functional foods and medicine. This study provides new insights into the extension of resource utilisation of *S. hexandrum*, as does the study by Pavlović et al. [102], indicating variable amounts of PCs in different organs of hawthorn (*Crataegus pentagyna* Willd.). While neochlorogenic and chlorogenic acids (5.2 and 4.85 mg∙g^−1^ DM, respectively) are the most abundant PCs in flowers, (-)-epicatechin and procyanidin B2 dominate in leaves and fruits. Studies of wild-growing plants of *Sambusus nigra* L. have shown that elderberry flowers are the most abundant in total polyphenols and phenolic acids, the leaf in flavonoids and the fruit in anthocyanins [103]. 

Plant organs have different functions and different levels of exposure to environmental conditions, which can affect the distribution of metabolites and antioxidant capacity. The concentration of PCs in a plant is closely related to the type and function of the organ in question. Andrade-Andrade et al. [104] identified the chemical profile of vanilla (*Vanilla planifolia* Jacks. ex Andrews) and determined the variable content of total phenolic compounds, total tannins, condensed tannins and flavonoids in leaves, stems, flowers and dried fruits. The concentration of PCs varied according to the function of the vanilla tissue. Leaves had the highest concentrations of flavonoids (127 mg∙100 g^−1^ DM) and condensed tannins (41 mg∙100 g^−1^ DM), protective compounds for tissues exposed to pathogens, herbivores and UV radiation. Vanilla fruit had the highest concentrations of total phenolic compounds and total tannins (749.6 and 102.1 mg∙100 g^−1^ DM, respectively). The leaves of *Aquilaria beccariana* are commonly used in Indonesia without consideration of leaf age. A study of young, mature and old leaves of *A. beccariana* by Anwar et al. [105] showed that leaf age affects the levels of secondary metabolites and their antioxidant capacity. The levels of PCs and flavonoids in mature leaves of *A. beccariana* were higher than those in young and old leaves. The levels of PCs in young and old leaves were higher than those of flavonoids, while the levels of PCs and flavonoids in mature leaves were not significantly different. The mature leaf extract had the highest antioxidant potential (IC_50_ 72.25 ± 0.72 ppm). Fluctuations in the concentration of metabolites may affect the level of antioxidant activity, thus influencing pharmacological effects. Alba et al. [106] showed that *Anredera cordifolia* (Ten.) Steenis leaf phenology and seasonality cause changes in the number of PCs and antioxidant activity levels. The total phenolic content (TPC) was highest in young leaves harvested in winter (54.4 GAE∙100 g^−1^ DM), and the highest antioxidant activity was found regardless of leaf phenology in leaves harvested in autumn (73.7%) and winter (71.1%). The authors found no significant correlation between TPC and antioxidant activity. Similar relationships may also apply to other plant organs harvested as wild medicinal plants. *Lythrum salicaria* L. (Lythraceae) has been known as a medicinal plant since ancient Greek and Roman times [107]. The official medicine is the whole flowering plant (*Lythri herba*—European Pharmacopoeia 6th edition), which is traditionally used to treat diarrhoea, chronic intestinal catarrh, haemorrhoids, eczema, varicose veins and bleeding gums [107,108]. Bencsik et al. [108] studied the chemical composition of twelve populations of *L. salicaria* (Hungary) and showed the highest flavonoid content in the leaves, followed by the flowering shoots. The contents of total polyphenols and tannins were higher in the tops of the flowering shoots than in the other organs. Environmental factors and leaf maturity determine the production of metabolites, and for these reasons it is essential to follow these changes to determine the best harvest time.

### 4.3. Development Stage

Plant phenols show considerable qualitative and quantitative variability at different genetic levels (between and within species and clones) and physiological and developmental stages [109]. Many herbaceous plants have peak levels of secondary metabolites, including PCs, at the flowering and fruiting stages [12,110]. The findings of Naghiloo et al. [110] on quantitative estimation of PCs’ content in roots and leaves of *Astragalus compactus* at different growth stages indicate that leaf and root extracts showed a significant increase in PCs during ontogeny, highest at the fruiting stage. Feduraev et al. [12] proved maximum PCs’ accumulation in generative organs during the flowering and fruiting of *Rumex crispus* and *R. obtusifolius*. The antioxidant activity of the tested parts of *R. crispus* and *R. obtusifolius* was arranged in the following order: generative part (flowers, seeds) > leaves > root > stem (for flowering and fruiting phases). Catechin accumulation studies showed variability in the distribution of the compounds throughout the plant. Both species presented significantly higher catechin levels in the first part of the root, which is closer to the ground during the flowering phase. Younger leaves, located closer to the generative part of the plant, contained more catechins than the others. On the other hand, the analysis of purple betony (*Stachys officinalis* L., Lamiaceae) herb by Bączek et al. [111] proved the highest tannin content during the vegetative development of the plant (in the second and third years of vegetation: 2.05% and 2.91%, respectively). The highest content of caffeic acid was found at the beginning of plant vegetation, while apigenin was found at full flowering and the beginning of seed setting. Bencsik et al. [108] showed that the total flavonoid content of the *Lythrum salicaria* L. herb was higher at full flowering (August) than at the beginning of flowering (July).

In the case of berry plants, in addition to the fruit, the leaves, inherently rich in antioxidants, are also harvested. Moreover, leaf extracts are now attracting increasing attention as products of nutraceutical importance. Lebedev et al. [93] reported that raspberry leaves harvested at three phenological phases contained more phenols (5.4 times) and flavonoids (4.1 times) and showed higher antioxidant activity (2.4 times in the FRAP assay, 2.2 times in the ABTS) than fruits. The fruit ripening stage was the optimal harvesting date of raspberry leaves for most of the cultivars studied. Pavlović et al. [102] reported seasonal variability in the amount and type of PCs and in the antioxidant activity of different organs of hawthorn (*Crataegus pentagyna* Willd.). The highest PCs’ content was found in leaves harvested at the early stages of maturity, with a decreasing trend at later stages. On the other hand, the fruit’s most noticeable changes were found in flavan-3-ol content. The highest content of (-)-epicatechin (21.1 mg∙g^−1^ FM) in the fruit was recorded in August, and procyanidins B2 and B5 were recorded in September (10.6 and 3.74 mg∙g^−1^ FM, respectively). A strong correlation was found between total PCs’ content and antioxidant activity. Variable amounts of PCs at different stages of development were also found in elderberry (*Sambucus nigra* L.) organs [103]. Leaves harvested before flowering contained the highest levels of phenolic acids and flavonoids, while those harvested at the beginning of flowering contained total polyphenols. The level of anthocyanins in this raw material was higher before and at the beginning of flowering than at the fruiting stage. In contrast, elderberry fruit harvested at full maturity contained more flavonoids than that harvested at the beginning of ripening.

Gemmotherapy is a therapeutic method that exploits the properties of extracts from fresh meristematic plant tissues, mainly buds and sprouts, by macerating them in ethanol and glycerol. Kovalikova et al. [30] showed that the harvesting time and location could significantly affect the chemical composition of buds. Changes in metabolite content concerning the sampling time may be related to the transition of buds from the dormant to the active growth phase, where metabolite requirements differ. In birch buds, the content of hydrolysing tannins (gallotannins and ellagitannins) and flavonoid aglycones decreased by up to 90%, depending on the transformation of buds into adult leaves. The content of phenolic acids (mainly hydroxycinnamic acid derivatives) increased.

On the basis of the studies cited above, it is difficult to clearly indicate the tendency of changes in the content of polyphenols during ontogenesis. The relationship between the age of the plant/age and type of organ and the content of phenolic compounds is quite complicated because it also depends on the specific genotype [105,106] and environmental conditions [31,106]. It should be emphasised, however, that the best possible knowledge of seasonal changes in polyphenol complex is important due to their biological activity [14,102].

### 4.4. Practical Aspects of Ontogenetic Variability

Knowing how the content of active substances changes during ontogeny is of great practical importance, as it is an essential factor in regulating the level of active ingredients in the raw material. It is also an indication for determining the optimum harvest date while considering possible genetic variability. Guajardo et al. [13] report that, according to traditional practices and ethnobotanical studies, the timing of harvesting the underground organs of *Valeriana carnosa* for medicinal purposes does not coincide with the seasons when the plants have the highest content of PCs. The authors switched the total PC content (TPC) of extracts from the roots and rhizomes of two *V. carnosa* populations at three phenological stages. The TPC ranged from 3.56 to 11.68 mg GAE∙g^−1^ DM in individual specimens. Eighteen PCs were identified, mainly phenolic acids, one of which (chlorogenic acid-isomer conjugate) was present only in the Hoya population at the vegetative stage and one (syringic acid) only in the Piltri population at the flowering and fruiting stages. PCs in the underground organs of *V. carnosa* varied qualitatively (between populations/stadia) and quantitatively at intra- and inter-population levels at different phenological stages. Qualitative and quantitative variation were found between individuals within each population and between populations. Bhandari et al. [89] showed changes in chemical composition, including broccoli florets’ total phenolic and flavonoid content and antioxidant activity at different developmental stages. Almost all cultivars had significantly higher contents of individual flavonoids (kaempferol, apigenin, quercetin and myricetin) at the intermediate and commercial stages than at the immature stage. The total PCs’ content increased with maturity progression in all genotypes, with the lowest and highest values occurring at the immature and commercial stages, respectively. Moreover, PCs’ content positively correlated with antioxidant activity.

## 5. Conclusions

From a medical and pharmaceutical point of view, PCs are among the best natural compounds, showing a range of health benefits. The excellent biological activity of PCs and their free radical and toxin scavenging properties point to new directions for medical research. A study of the literature reveals that wild and cultivated plants for various purposes are rich natural sources of PCs of different natures and health-promoting effects. Agro-industrial product derivatives, rich in active compounds and with significant health-promoting benefits, are also worthy of consideration. Green alternatives that reduce environmental pollution and waste generation include renewable biomaterials, including by-products of cereal, vegetable and fruit processing. In recent years, innovative applications of agro-industrial by-products have contributed to the continued development of the bioeconomy and biotechnology through the extraction or valorisation of natural PCs, or both.

Phytochemical and biological plant research is becoming a leading activity of interdisciplinary research teams. The aim of this research is, among other things, to identify opportunities for the production of natural phytotherapeutics, biocosmetics and preservatives. Plant polyphenols are stress metabolites. When analysing their content and potential health-promoting activities, these compounds’ role in the plant’s life should be taken into account, as well as their possible variability. These relationships should be considered at the genome/organism/phase of development level under specific environmental conditions. Experiments on potential sources of polyphenols are fundamental and timely due to the broad and potent biological activity of these components. The quantitative and qualitative evaluation of the PCs’ complex of various wild and cultivated plant species, in conjunction with the antioxidant activity of the extracts obtained from them, taking into account various causes of variability, including genetic and ontogenetic variability, is a good research direction. In addition, further studies are warranted to analyse PCs’ potential activities and determine their efficacy and safety profile for therapeutic purposes.

## Figures and Tables

**Table 2 molecules-28-01731-t002:** Phenolic composition of some cereal crops.

Cereals	Compounds	Ref.
Barley (*Hordeum vulgare* L.)	flavonoids, proanthocyanidins	[32,35,36]
Buckwheat (*Fagopyrum esculentum* Moench)	flavonoids (rutin, catechin), proanthocyanidins	[32,35]
Emmer (*Triticum dicoccoides* (Körn. ex Asch. & Graebn.) Schweinf.)	polyphenols	[32]
Einkorn (*Triticum monococcum* L.)	polyphenols	[32]
Maize (*Zea mays* L.)	anthocyanins, flavonoids (kaempferol, morin, naringenin, quercetin rutin), phenolic acids (caffeic acid, chlorogenic acid, ferulic acid)	[32,37]
Millets (*Panicum miliaceum* L.)	anthocyanins, flavonoids, phenolic acids	[32,38]
Oats (*Avena sativa* L.)	polyphenols	[32,35]
Rice (*Oryza sativa* L.)	anthocyanins, phenolic acids, proanthocyanidins	[32,35]
Rye (*Secale cereale* L.)	anthocyanins, phenolic acids	[32]
Sorghum (*Sorghum bicolor* (L.) Moench)	anthocyanins, flavonoids, phenolic acids (chlorogenic acid, gallic acid, ferulic acid), proanthocyanidins	[32,34,38]
Triticale (x *Triticosecale* spp.)	phenolic acids, proanthocyanidins	[32]
Wheat (*Triticum aestivum* L.)	anthocyanins, flavonoids, phenolic acids (hydroxycinnamic and hydroxybenzoic)	[32,35]
Wheat (*Triticum durum* Desf.)	anthocyanins, flavonoids, phenolic acids, polyphenols	[32,36]

**Table 3 molecules-28-01731-t003:** Examples of phenolic compounds obtained from plant cell, tissue and organ cultures.

Compound	Source	Ref.
Arbutin	*Catharanthus roseus* L.	[43]
Caffeic acid	*Phoenix dactylifera* L.; *Polyscias filicifolia* Bailey	[50,51]
Cambodianol	*Dracaena cambodiana* Pierre ex Gagnep.	[44]
Chlorogenic acid	*Cynara cardunculus* L.; *P. filicifolia* Bailey; *Spiraea betulifolia* ssp. *aemiliana* (C.K. Schneid.) H. Hara	[51,52,53]
Cinnamic acid	*S. betulifolia* ssp. *aemiliana* (C.K. Schneid.) H. Hara, *Iris pseudacorus* L.	[53,54]
Cynarin	*C. cardunculus* L.	[52]
7,4′-Dihydroxyflavanone	*Dracaena cambodiana* Pierre ex Gagnep.	[44]
Ferulic acid	*C. cardunculus* L.; *P. filicifolia* Bailey	[51,52]
Iristorigenin A	*I. pseudacorus* L.	[54]
Isochlorogenic acid A	*Tanacetum vulgare* L.	[55]
Kaempferol	*Alpinia zerumbet* (Pers.) Burtt et Smith; *Ginkgo biloba* L.; *P. dactylifera* L.; *S. betulifolia* ssp. *aemiliana* (C.K. Schneid.) H. Hara	[43,46,50,53]
Lavandoside	*I. pseudacorus* L.	[54]
Myricetin	*Drosera aliciae* R. Hamet	[56]
p-Coumaric acid	*C. cardunculus* L.; *S. betulifolia* ssp. *aemiliana* (C.K. Schneid.) H. Hara	[52,53]
Quercetin	*D. aliciae* R. Hamet; *S. betulifolia* ssp. *aemiliana* (C.K. Schneid.) H. Hara	[53,56]
Resveratrol	*Vitis vinifera* L.	[43]
Rosmarinic acid	*Anchusa officinalis* L.; *Duboisia* R. Br. spp.; *Hyssopus officinalis* L.; *Lavandula officinalis* Chaix., *Ocimum basilicum* L.; *Salvia officinalis* L.	[43,47,57]
Rutin	*Alpinia zerumbet* (Pers.) Burtt et Smith	[46]
Tectoridin	*I. pseudacorus* L.	[54]
Tectorigenin	*I. pseudacorus* L.	[54]

**Table 4 molecules-28-01731-t004:** Phenolic components found in some agri-food by-products.

Source	Compound	Ref.
Apple pomace	chlorogenic acid, epicatechin, phloridzin	[66]
Apple peels	caffeic acid, caffeic acid-4-*O*-glucoside, 5-caffeoylquinic acid, 3-caffeoylquinic acid, catechin 3-*O*-glucose, cyanidin 3-*O*-arabinoside, 3,4-dicaffeoylquinic acid, epicatechin, malvidin 3-*O*-(6′-*p*-coumaroyl)-glucoside, malvidin 3-*O*-glucoside, peonidin 3-*O*-glucoside, petunidin 3-*O*-(6′-*p*coumaroyl)-glucoside, phloridzin, phloretin, quercitrin, quercetin, rutin	[64]
Blackcurrant buds	flavonol glycosides quercetin, myricetin, isoramnetin, kaempferol	[67]
Blackcurrant leaves	chlorogenic acid, kaempferol-3-*O*-rutinoside, neochlorogenic acid	[67]
Cereal bran	cyanidin-3-glucoside, ferulic acid	[66]
Cottonseed	gallic acid, quercetin, flavonol glycosides, 3,4-dihydroxybenzoic acid	[16]
Hemp seeds	cinnamic acid, ferulic acid, p-coumaric acid, p-hydroxybenzoic acid, protocatechuic acid, syringic acid, vanillic acid,	[68]
Olive pomace	caffeic acid, catechol, elenolic acid, hydroxytyrosol, oleuropein, p-coumaric acid, vanillic acidrutin, verbascoside, tyrosol	[69,70]
Tomato by-products	caffeic acid, catechin, gallic acid, naringenin, procatchoic acid, quercetin, rutin, vanillic acid	[64,66]

**Table 5 molecules-28-01731-t005:** Examples of plant genotypes with variable levels of phenols.

Plant Genotype	Compound	Ref.
*Brassica oleracea* L. var. *italic* Plenck	apigenin, kaempferol, myricetin, quercetin	[89]
*Fragaria* x *ananassa* Duch.	benzoic acid derivatives, ellagtannins, flavons, flavonols, phenylpropanoids	[92]
*Origanum vulgare* L.	chicory acid, coumarin, luteolin, quercetin, rosmarinic acid	[91]
*Trigonella* L., *Medicago* L.	apigenin, caffeic acid, chlorogenic acid, *p*-coumaric acid, ellagic acid, ferulic acid, gallic acid, kaempferol, quercetin, rosmarinic acid, syringic acid	[91]
*Vaccinium macrocarpon* Aiton	caffeic acid, catechin, chlorogenic acid, p-coumaric acid, epicatechin, ferulic acid, gallic acid, hyperoside, kaempferol, neochlorogenic acid, quercetin, phloretin, phloridzin, procyanidin A2, rutin, vanillic acid	[27]

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
