# Peer review of "Phenolic Compounds from New Natural Sources—Plant Genotype and Ontogenetic Variation"

_molecules, 2023, doi:10.3390/molecules28041731_

Round 1

Reviewer 1 Report

1) Abbreviations should be explained as soon as they appear in the text such as: GAE in page 5 line 195, QE and DM page 5 line 201, DPPH in page 5 line 204, CAT, TPX and SOD in page 5 line 231, FM in page 6 line 257

2) several sentences require the addition of references:

- Phenolic compounds (PCs) are essential components of food. In addition to their 28 strong antioxidant properties, they influence the sensory characteristics of food products.

- PCs are stress metabolites pro-31 duced by plants to protect themselves against abiotic and biotic factors

- Two main pathways 38 are involved in the biosynthesis of PCs, the shikimic acid pathway and the malonic acid 39 pathway, with the former being the main pathway in plants.

- Polyphenols are isolated and purified from plants (fruits, vegetables and 80 agricultural byproducts) and converted into medicines and supplements

- The plant occurs naturally in North America's 132 eastern and central regions but is increasingly cultivated in Europe and other continents.

- Wild plants represent a valuable genetic resource that can be used in breeding 169 programmes to increase the resistance of crop plants and improve their nutritional and 170 pharmacological value.

- In addition to naturally occurring plants, cultivated plants, such as cereals, 174 are also an interesting source of polyphenols and are increasingly being studied for their 175 PCs and antioxidant activity.

- Cereals are an essential food category 179 for many of the world's populations, with an annual production of more than 2,700 tonnes 180 when supply and demand are balanced.

- Sorghum [Sorghum bicolor (L.) Moench] and pearl millet [Pennisetum glaucum (L.) R. 186 Br.] are cereal crops from the Poaceae family cultivated in various regions of the world.

- The above study found higher amounts of PCs in 205 sorghum than in pearl millet. Polyphenols are the main bioactive compounds of sorghum 206 and are present in all varieties of this cereal crop. It should be added that polyphenols and 207 phytates in sorghum and millet grains are also known to be anti-nutritional factors, as 208 they form insoluble complexes with minerals such as iron, zinc and calcium, reducing 209 their bioavailability.

- Many plants containing high-value PCs are 247 difficult to cultivate on a large scale due to specific environmental requirements

- Suitable 248 plant cell cultures have become an attractive alternative PCs source, especially when nat-249 ural resources are limited. The use of plant cell cultures can be advantageous because they 250 grow under strictly controlled conditions, allowing the easy addition of hormones, bio-251 synthetic precursors and other compounds. Additionally, these methods are often more 252 efficient and cost-effective than other methods of obtaining active substances.

- Food processing causes significant nutrient losses, and waste generation causes se-304 vere economic and environmental problems

- Fresh fruits and vegetables represent a significant segment in the functional and nu-311 tritional food sector. In fruit and vegetable production and processing, unused parts of 312 the plants remain, often containing significant amounts of bioingredients. Fruit and veg-313 etables include hulls, skins, pods, pomace, seeds and stems, which are usually discarded 314 despite containing potentially beneficial compounds such as carotenoids, dietary fibre, 315 enzymes, and polyphenols.

Author Response

  1. Thank you very much for your valuable comments. The article has been corrected, taking into account the comments and suggestions of the Reviewer.

Abbreviations should be explained as soon as they appear in the text such as: GAE in page 5 line 195, QE and DM page 5 line 201, DPPH in page 5 line 204, CAT, TPX and SOD in page 5 line 231, FM in page 6 line 257 - Abbreviations have been explained or full names have been entered

2) several sentences require the addition of references: - Appropriate literature has been added to all sentences

Reviewer 2 Report

Dear,

Although the review about phenolic compounds is detailed, I believe that in this form it is not acceptable for publication and must go through a major revision.

1) The work is not well conceptualized. Namely, there are a various sub-topics that are difficult to follow, and often it goes from topic to topic and the areas are mixed up, i.e. are overlapping, especially in part 4. Ontogenetic variation, which is not clearly written so presented data and text are not easy to follow (for example part of subsection 4.5., lines 693-697, should be part of 4.2; and part of 4.5, lines 687-693, is more appropriate for subsection 4.4.

2) Also, throughout the paper, the antioxidant activity is emphasized, which is not actually the topic of this paper and represents the generally known activity of phenolic compounds. Unexpectedly at one point (lines 290-302) there is a whole paragraph about the potential use in cancer therapy, and in the part (lines 340-346) a section about the antihyperglycaemic and antihyperlipidaemic activity of a certain phenol, specifically phloridzin, is added. Similarly to this paragraph (341-345), the text often gives more attention to individual phenols without explaining the connection with the topic itself, but as presenting the rest of the results from the cited work (296-302; 405-408; 515-524; 646-648; etc.). If the author wanted to indicate the importance of phenol compounds within the text, he/she would have to do it in a more systematic way. Written this way, the manuscript presents only a lot of data and information without any clear meaning and implication about the meaning and bioactivity of phenolic compounds. In this regard, the word UNUSUAL (line 10) should be deleted in the Abstract, because the bioactivities of PCs are not unusual. In addition in the abstract (line 17) the extraction techniques are mentioned, but they are not properly discussed in the entire text (although ”ethyl acetate fraction” is mentioned at one place (line 119).

3. It is mandatory for the author to correct Figure 1: to delete the structure of phenolic acids (at the top right), to add “hydroxy” before cinnamic acids, to check the structures of other compounds.

4. Scientific terms are often confused throughout the text. For example, in the Abstract, tissue culture is mentioned (line 13), in the title the subtitle 2.3 Plant cell culture (line 243), and then in the rest of the text again plant tissue culture (244) or cell culture (line 246). Specifically, for this subtitle, it would be better to put Plant cell, tissue and organ cultures because, in addition to cell cultures, the text also discusses tissue culture and organ cultures (line 264-270). Considering that this manuscript is a review, I strongly recommend to the authors to provide a tabular overview of the data, where, for example, they would provide an overview of the established in vitro cultures, indicating which phenolic compounds are produced. The same recommendation applies to the all other parts of Manuscript, as it will enable a clearer overview of the data and information.

Additionally, regarding this subtopic, the author in the section 5. Conclusions, states “An interesting potential source of phenols appears to be the plant cell cultures, which extract active substances more efficiently and cost-effectively than other methods”, which is not supported by any reference, and should be with at least a few, specifically to show that plant cell cultures are cost-effectively and more efficiently than other methods. In addition to the lack of references, the sentence itself should be corrected so that it has the correct meaning.

Other suggestions, such as italic font, typos, etc., I did not note because it is first and foremost necessary to change these main things abovementioned.

Author Response

Thank you very much for your valuable comments. Preparing a review paper is not easy, especially with a broad, multithreaded topic. However, the most important thing is the transparency of the work, hence every comment of the reviewer is very helpful and has been thoroughly analyzed. The article has been corrected, taking into account the comments and suggestions of the Reviewer.

1) The work is not well conceptualized. Namely, there are a various sub-topics that are difficult to follow, and often it goes from topic to topic and the areas are mixed up, i.e. are overlapping, especially in part 4. Ontogenetic variation, which is not clearly written so presented data and text are not easy to follow (for example part of subsection 4.5., lines 693-697, should be part of 4.2; and part of 4.5, lines 687-693, is more appropriate for subsection 4.4. [The chapter has been revised according to the reviewer's comments. The fragments suggested by the Reviewer have been moved to the appropriate place. The table suggested in the review has been added]

2) Also, throughout the paper, the antioxidant activity is emphasized, which is not actually the topic of this paper and represents the generally known activity of phenolic compounds. Unexpectedly at one point (lines 290-302) there is a whole paragraph about the potential use in cancer therapy [Explanation was added], and in the part (lines 340-346) a section about the antihyperglycaemic and antihyperlipidaemic activity of a certain phenol, specifically phloridzin, is added. [This part has been removed]. Similarly to this paragraph (341-345), the text often gives more attention to individual phenols without explaining the connection with the topic itself, but as presenting the rest of the results from the cited work (296-302; 405-408; 515-524; 646-648; etc.). If the author wanted to indicate the importance of phenol compounds within the text, he/she would have to do it in a more systematic way. Written this way, the manuscript presents only a lot of data and information without any clear meaning and implication about the meaning and bioactivity of phenolic compounds. In this regard, the word UNUSUAL (line 10) should be deleted in the Abstract, because the bioactivities of PCs are not unusual. In addition in the abstract (line 17) the extraction techniques are mentioned, but they are not properly discussed in the entire text (although ”ethyl acetate fraction” is mentioned at one place (line 119). [The word UNUSUAL  was removed, tables was added and the indicated fragments of the abstract have been corrected]

  1. It is mandatory for the author to correct Figure 1: to delete the structure of phenolic acids (at the top right), to add “hydroxy” before cinnamic acids, to check the structures of other compounds. [Figure has been removed]
  2. Scientific terms are often confused throughout the text. For example, in the Abstract, tissue culture is mentioned (line 13), in the title the subtitle 2.3 Plant cell culture (line 243), and then in the rest of the text again plant tissue culture (244) or cell culture (line 246). Specifically, for this subtitle, it would be better to put Plant cell, tissue and organ cultures because, in addition to cell cultures, the text also discusses tissue culture and organ cultures (line 264-270). Considering that this manuscript is a review, I strongly recommend to the authors to provide a tabular overview of the data, where, for example, they would provide an overview of the established in vitro cultures, indicating which phenolic compounds are produced. The same recommendation applies to the all other parts of Manuscript, as it will enable a clearer overview of the data and information. [Section title was changed, tables was added and the indicated fragments of the abstract have been corrected]

Additionally, regarding this subtopic, the author in the section 5. Conclusions, states “An interesting potential source of phenols appears to be the plant cell cultures, which extract active substances more efficiently and cost-effectively than other methods”, which is not supported by any reference, and should be with at least a few, specifically to show that plant cell cultures are cost-effectively and more efficiently than other methods. In addition to the lack of references, the sentence itself should be corrected so that it has the correct meaning. [The indicated sentence has been deleted.]

Other suggestions, such as italic font, typos, etc., I did not note because it is first and foremost necessary to change these main things abovementioned.

Round 2

Reviewer 2 Report

Dear,

All suggestions are adopted correctly by author. The modified and improved Manuscript is now ready for publication.

 All the best.